# Complete Chloroplast Genome Sequences of Four Species in the *Caladium* Genus: Comparative and Phylogenetic Analyses

**DOI:** 10.3390/genes13122180

**Published:** 2022-11-22

**Authors:** Yuanjun Ye, Jinmei Liu, Yiwei Zhou, Genfa Zhu, Jianjun Tan, Yechun Xu

**Affiliations:** 1Guangdong Provincial Key Lab of Ornamental Plant Germplasm Innovation and Utilization, Environmental Horticulture Research Institute, Guangdong Academy of Agricultural Sciences, Guangzhou 510640, China; 2Key Laboratory of Urban Agriculture in South China, Ministry of Agriculture, Guangzhou 510640, China

**Keywords:** *Caladium*, ornamental foliage plant, chloroplast genome, natural selection pressure, phylogenetic analysis

## Abstract

Caladiums are promising colorful foliage plants due to their dazzling colors of the leaves, veins, stripes, and patches, which are often cultivated in pots or gardens as decorations. Four wild species, including *C. bicolor*, C. *humboldtii*, *C. praetermissum,* and *C. lindenii,* were employed in this study, where their chloroplast (cp) genomes were sequenced, assembled, and annotated via high-throughput sequencing. The whole cp genome size ranged from 162,776 bp to 168,888 bp, and the GC contents ranged from 35.09% to 35.91%. Compared with the single large copy (LSC) and single small copy (SSC) regions, more conserved sequences were identified in the inverted repeat regions (IR). We further analyzed the different region borders of nine species of Araceae and found the expansion or contraction of IR/SSC regions might account for the cp genome size variation. Totally, 131 genes were annotated in the cp genomes, including 86 protein-coding genes (PCGs), 37 tRNAs, and eight rRNAs. The effective number of codons (ENC) values and neutrality plot analyses provided the foundation that the natural selection pressure could greatly affect the codon preference. The GC_3_ content was significantly lower than that of GC_1_ and GC_2_, and codons ending with A/U had higher usage preferences. Finally, we conducted phylogenetic relationship analysis based on the chloroplast genomes of twelve species of Araceae, in which *C. bicolor* and *C. humboldtii* were grouped together, and *C. lindenii* was furthest from the other three *Caladium* species occupying a separate branch. These results will provide a basis for the identification, development, and utilization of *Caladium* germplasm.

## 1. Introduction

The genus *Caladium* Vent. (family Araceae) includes perennial herbs native to the tropical regions of South and Central America [1,2,3]. Most *Caladium* species are distributed in the Amazon rainforest either in open areas or beside streams [4,5]. *Caladium* spp. is regarded as the most promising colorful foliage plant with great variation in leaf color [6,7]. Because of the dazzling colors of the leaves, veins, stripes, and patches, as well as the long ornamental period, they are often cultivated in pots or gardens as decorations [8]. The ornamental effect for urban areas is excellent, and hence this genus is collectively known as the “Queen of Foliage Plants” [9,10]. In many countries of Europe and America, *Caladium* is grown as a replacement for traditional grasses and flowers for the purposes of arranging flower beds and flower borders or creating a unique landscape effect of “no flowers is better than flowers” [11,12,13]. In recent years, *Caladium* has become a new favorite foliage plant due to its colorful leaves, short production cycle, and high selling profit which conferred its high popularity in domestic and foreign markets [14,15].

Over nearly 150 years of selection and breeding, more than 2000 varieties have been cultivated [16,17,18]. At present, there are more than 90 varieties on the market, among which more than 50 varieties are grown for large-scale flower production [19]. Since the beginning of this century, China has successively introduced some varieties of colorful *Caladium* as ornamental plants, which are greatly appreciated by retailers and young people [20].

The number of native species of the genus *Caladium* has always been controversial, ranging from 7 to 17. It has been reported earlier that *C. marmoratum*, *C. picturatum,* and *C. steudnirifolium* should be more appropriately merged into *C. bicolor* [21]. However, Croat recommended keeping the three species separate [22]. In 2013, *C. clavatum* and *C. praetermissum* were added to the list of species of wild *Caladium*, jointly released by Kew Gardens in the UK and the Missouri Botanical Garden in the US [23]. This increased the number of native species of *Caladium* to 14. Cao and Deng investigated the genome size and chromosome numbers of ten native species and 53 cultivars of *Caladium* [24]. They found that there are two genome types, thus providing some evidence for the evolutionary origin of *Caladium* germplasm.

Genetic and kinship analyses of *Caladium* germplasm have also received some attention. Loh et al. used 17 pairs of amplified fragment length polymorphism (AFLP) markers to accurately distinguish two wild species (*C. bicolor* and *C. schomburgkii*) and six cultivars of *Caladium* [25]. Deng et al. used simple sequence repeat (SSR) markers to analyze the genetic relationships among 45 major cultivars and seven wild species of *Caladium* [23]. The authors reported that the genetic variation rate among cultivars was low (44.4%). However, the two wild species *C. bicolor* and *C. schomburgkii,* exhibited high genetic consistencies with the cultivars, whereas the genetic consistencies between other wild species and cultivars were low. Moreover, *C. steudnirifolium* and *C. lindenii* were less genetically related to other wild species.

The chloroplast is the site of photosynthetic activity and is also one of the most important organelles of green plants, playing an important role in the inheritance and expression of plant genetic material [26,27,28,29]. The chloroplast has its own genome with an independent genetic system consisting of multiple copies of circular DNA molecules (110–210 kb), although few plant species have multiple linear copies of chromosomes [30,31,32]. The chloroplast genome has a typical tetrad structure featuring a large single-copy region, a small single-copy region, and a pair of inverted repeats (IR) regions [33,34]. The differences among plant species are attributed to the expansion and contraction of the IR regions [35]. Compared with the nuclear genome, the chloroplast genome shows maternal uni-parental inheritance, highly conserved structure, and less gene recombination [36,37,38]. Therefore, the chloroplast genome sequence is widely used in plant systematics, phylogeny, and population genomics [39,40].

Next-generation sequencing (NGS) technology has been widely used to identify functional genes and DNA markers massively due to its high throughput, high accuracy, and low cost. With the recent improvement of sequencing technology, the ongoing upgrading of sequencing platforms, and the development of a series of dynamic assembly and annotation software such as plasmaSPAdes, NOVOPlasty, and GetOrganelle, the chloroplast genomes of more than 3000 plant species have been sequenced and analyzed [41,42,43,44]. Remarkably, the chloroplast genome of *Caladium* has not yet been sequenced, and few phylogenetic studies based on chloroplast genomics have been reported.

In this study, four wild species of *Caladium* were used, and their chloroplast genomes were sequenced, assembled, and annotated via high-throughput sequencing. The characteristics of IR boundary and spacer loci in the chloroplast genomes were analyzed, and their genetic diversity and phylogenetic relationship were investigated. These results will provide a basis for the identification, development, and utilization of *Caladium* germplasm.

## 2. Results

### 2.1. The Structure of the Chloroplast Genomes of the Four Caladium Species

The chloroplast genome sizes of *C. bicolor* and *C. humboldtii* showed high similarity (162,933 bp and 162,776 bp, respectively) with a 157 bp difference, whereas those of *C. praetermissum* and *C. lindenii* were slightly larger (165,286 bp and 168,888 bp, respectively) (Figure 1). The GC contents of the four cp genomes were 35.87%, 35.91%, 35.64%, and 35.09%, respectively (Table 1). The cpDNA of the four samples showed a typical tetrad ring structure, consisting of a pair of inverted repeat regions (IRa and IRb; 26,277, 26,277, 26,484 and 26,472 bp in length, respectively), a large single copy region (LSC; 89,209, 88,986, 91,168 and 93,162 bp in length, respectively) and a small single copy region (SSC; 21,170, 21,236, 21,150 and 22,782 bp in length, respectively). Table 1 also shows that the LSC region of *C. humboldtii* was also significantly shorter than that of the other species, whereas that of *C. lindenii* was the longest. In terms of SSC length, *Zamioculcas zamiifolia* showed the smallest, but *C. lindenii* had the greatest SSC. For IR size, *Z. amazonica* exhibited the shortest while *Z. zamiifolia* had the longest IR. Moreover, we compared the differences in chloroplast genome sequences between the ON7070731 and NC_060474 (Appendix A). Results showed there were only some variations in the noncoding region and one SNP in the coding region, which did not affect the genetic structure.

A total of 131 genes were identified in the chloroplast genome of *Caladium*, including 86 protein-coding genes (PCGs), 37 tRNAs, and eight rRNAs (Appendix A). Most genes appeared in the LSC or SSC region in single copy form. Among them, 12 genes were assigned to the SSC region, including 11 PCGs (*ndhF*, *rpl32*, *ccsA*, *ndhD*, *psaC*, *ndhE*, *ndhG*, *ndhI*, *ndhA*, *ndhH,* and *rps15*) and one tRNA (*trnL-UAG*). There were 83 genes in the LSC region, including 61 PCGs and 22 tRNAs. Only 16 genes were detected in the IR region, including five PCGs (*rpl2*, *rpl23*, *ycf2*, *ndhB,* and *rps7*), seven tRNAs (*trnI-CAU*, *trnL-CAA*, *trnV-GAC*, *trnI-GAU*, *trnA-UGC*, *trnI-ACG,* and *trnA-GUU*) and four rRNAs (*rrn16*, *rrn23*, *rrn4.5* and *rrn5*). The *ycf1* gene spanned the SSC region and the IR region. The *rps12* had two copies, each having three exons, and the two copies shared the first exon, which was in the LSC region, while the other two exons were in the IR region.

### 2.2. Analysis of Contraction and Expansion of the IR Region

By comparing the gene distribution of IR/LSC and IR/SSC border regions in the chloroplast genomes of four *Caladium* species and those of related species, the expansion or contraction of IR/SSC boundary regions was assessed. As shown in Figure 2, the nine sequences had similar gene structures and sequences and the genes distributed near the boundaries of IR/LSC and IR/SSC were *rps19*, *rpl22*, *rpl2*, *ycf1*, *trnH,* and *psbA*. Among them, the IR boundaries of *C. bicolor* and *C. humboldtii* were identical, whereas *C. praetermissum* differed only in the SSC/IRa boundary. In *C. bicolor* and *C. humboldtii*, the sizes of the *ycf1* gene were 5241 and 422 bp in the SSC and IRa regions, respectively, while these sizes in *C. praetermissum* were 5292 bp and 422 bp, respectively. As for *C. lindenii*, it was quite different from the other three *Caladium* species, and its boundaries were different, indicating that this species may have undergone a unique evolutionary process.

### 2.3. GView Analysis

To gain a deeper understanding of the phylogenetic relationships among the different species of *Caladium* and the differences from other closely related species, we used the GView tool to create a circle map of the chloroplast genomes with the assembled *C. bicolor* genome as a reference. The characteristics and structural variation of all chloroplast genomes were evaluated. Figure 3 shows that the nine genomes had similar structures. Compared with the IR region, the LSC and SSC regions varied greatly among different species. Even within the same genus, the chloroplast genomes of the studied four species of *Caladium* showed inconsistencies. Particularly, there existed tiny variations among the intergenic regions. In terms of the genome structures, *C. bicolor* and *C. humboldtii* were identical. However, the cp genome of *C. praetermissum* was similar to that of *Xanthosoma sagittifolium.* Moreover, the cp genome of *C. lindenii* was similar to that of *Syngonium angustatum*.

### 2.4. Analysis of Chloroplast Microsatellites and Repeat Sequences

In this study, we analyzed the distribution of SSRs in the cp genomes of four *Caladium* species. As shown in Figure 4, there were only three types of SSRs, including single-base, two-base, and three-base repeats. Three-base repeats were only present in *C. lindenii* but not in the other three species. Most SSRs were single-base repeats, accounting for 63.0–81.7% of the total SSRs, which was more than all other repeat types combined. The total numbers of SSRs in *C. bicolor* and *C. humboldtii* were similar (91 and 89, respectively). *C. praetermissum* had the least number of SSRs, whereas *C. lindenii* had the highest. The SSRs of the four *Caladium* genomes were mostly distributed in the LSC region and the non-coding region, compared to other regions. In terms of repeating units, A/T repeats were significantly more than C/G repeats. All two-base and three-base repeats were AT/AT repeats and AAT/ATT, respectively, but there were no other types of repeats.

### 2.5. Analysis of Selection Pressure and Codon Bias

The relative synonymous codon usage (RSCU) tool was used to evaluate the use of synonymous codons in coding regions, where a larger RSCU indicated a stronger bias. Our data showed that the content of leucine was highest in the chloroplast genomes, followed by serine and arginine, whereas the number of codons of tryptophan was the least. As shown in Figure 5, all amino acids except tryptophan used two or more synonymous codons. For example, isoleucine was encoded by three synonymous codons (alanine, glycine, and proline). Threonine and valine were encoded by four synonymous codons, and leucine, serine, and arginine were encoded by six synonymous codons. There were 32 RSCU values greater than one, of which 13 of them ended with A and 16 end with U. These findings were consistent with previous studies, which showed that codons ending with A/U in plants had higher usage preference. As shown in Appendix A, the codon preferences of different *Caladium* species showed high conservation, where two codons exhibited the consistent RSCU value (AUG, 1.997; GUG, 0.003) and represented the extreme value in the four *Caladium* species. However, there were still some discrepancies among different materials, which mainly focused on the number of codons. For most codons, the codon preferences of *C. bicolor* and *C. humboldtii* were almost identical, significantly differing from those of the other two species, especially in *C. lindenii*.

In order to analyze the trend of codon usage bias in the cp genomes of nine species, the values of the effective number of codons (ENC) were investigated. As in Appendix A, the ENC values varied from 17.158 to 61.000, showing different extents of codon preferences among the species and indicating that the codon preference was weak. To further explore the details, the distribution of the ENC values of the coding genes in the genomes was exhibited in Appendix A.

GC_1-3_ means the GC content of three different positions of each codon. The overall GC content of the cp genomes varied among the nine species and ranged from 29.06% to 46.21% (Appendix A). As expected, the GC_1_, GC_2_, and GC_3_ contents varied significantly across species and also among genes in the genomes. The average value of GC_3_ in the cp genomes was 28.63%, and the GC_1_ and GC_2_ were 46.07%, and 40%, respectively. The GC_3_ content was significantly lower than that of GC_1_ and GC_2_. We also found that the greatest difference in GC content existed in GC_3_, which was widely applied to better illustrate the codon usage variation. The neutrality plot was shown in Appendix A, which revealed little correlation between GC_3_ and GC_12_. These results provided the foundation that the natural selection pressure could greatly affect codon preference. 

### 2.6. Phylogenetic Relationship Analysis

We selected the chloroplast genomes of twelve species of Araceae to explore the genetic relationship between *Caladium* and its relatives. The phylogenetic tree was constructed based on the complete chloroplast genome sequences using maximum likelihood (ML) and Bayesian inference (BI) methods. The results showed that the topological structures of the ML and BI analyses were identical and that most clades had high posterior probabilities and bootstrap values. As shown in Figure 6, four *Caladium* species were divided into different branches, in which two *C. bicolor* (ON7070731 and NC_060474) and *C. humboldtii* were grouped together, with *Z. amazonica* being relatively close. Furthermore, *C. praetermissum* and *X. sagittifolium* were categorized into a branch, while *C. lindenii* was furthest from the other three *Caladium* species occupying a separate branch and being relatively closely related to *Syngonium angustatum*. These findings are consistent with previous reports, in which *C. lindenii* was later classified as *Caladium* genus, and its appearance and resistance were more similar to those of *S. angustatum*.

## 3. Discussion

The cp genomes of the four *Caladium* species all had the typical circular tetrad structure of angiosperm cp genomes and were quite different in length. In terms of genome sequence length and number of annotated genes, except for *Z. amazonica*, which has 130 genes, the other *Caladium* species, as well as the related species, had 131 genes, indicating that the cp genome of *Caladium* had a certain degree of conservation. The GC content was reported as an important indicator for judging the genetic relationship among species [45,46]. In our study, the gene types, numbers, and order of genes encoded by the genomes of the four *Caladium* species were identical, with highly similar G + C content. Based on the G + C content in the region sequence, the rank from high to low was IR, LSC, and SSC, which is a ubiquitous phenomenon in many plant species [47,48,49]. The IR boundary was different among different species, and the fluctuation of the IR boundary was the main reason for the difference [50]. The cp genomes of *C. bicolor* and *C. humboldtii* showed the smallest difference in the IR regions among those of the four *Caladium* species. There was no significant difference in the contraction and expansion of the IR regions between the two. Therefore, the GC content and the IR region boundary conditions, to a large extent, indicate that *C. bicolor* and *C. humboldtii* are very closely related but are relatively distinct from *C. humboldtii* and *C. lindenii*.

Codon bias affects translation initiation, elongation, and accuracy, as well as mRNA splicing and protein folding [51,52,53]. Therefore, codon preference can also reflect kinship to a certain extent. Among the chloroplast genes of the four *Caladium* species, the number of codons of leucine was the largest, the number of tryptophan occurrences was the least, and codons ending in A/T were preferred. This feature is consistent with that of most plant species [54,55]. The codon preferences of *C. bicolor* and *C. humboldtii* showed almost no difference, suggesting that the two species are closely related, while the codon preferences of *C. humboldtii* and *C. lindenii* were markedly different, indicating that they may have unique evolutionary positions.

Microsatellite sequences (microsatellite DNA), also known as SSR, are 1–6 bp repeats that are widely distributed in the cp genome. SSR is highly polymorphic and specific and is a valuable marker for studying gene flow, population genetics, and genetic mapping [56]. Except for *C. lindenii*, the repeat types and distribution numbers of the other three *Caladium* species were roughly the same. Identification of these SSR loci can provide candidate molecular markers for research on the genetic diversity and conservation genetics of *Caladium*. The nodes in the LSC, SSC, and IR regions of the four *Caladium* species were highly conserved, indicating that the cp genome structure of *Caladium* is highly conserved. Phylogenetic analysis classified *C. lindenii* as a single clade that was far from the other three *Caladium* species, which is consistent with previous reports [23,24,25]. As for the limited research on the phylogenetic analysis, AFLP and SSR markers were often used to identify their kinship among *Caladium* cultivars. In this paper, we provided more accurate findings on the phylogenetic relationship analysis among different species of Caladieae. Based on the above results, we conclude that *C. lindenii* has greater specificities in cp genome structure, IR region contraction and expansion, SSR distribution, and codon preference, being relatively close to *S. angustatum*. Therefore, this species is suggested to be classified into the genus *Synaptocarpus*.

## 4. Materials and Methods

### 4.1. Plant Materials

Plants of four *Caladium* species were collected from the Environmental Horticulture Research Institute of Guangdong Academy of Agricultural Sciences (23° 23′ N, 113° 26′ E), namely, *C. bicolor*, *C. humboldtii* ‘Mini White’, *C. praetermissum* ‘Hilo Beauty,’ and *C. lindenii*. The phenotypic characteristics are shown in Figure 7. Young leaves were collected and rinsed thoroughly with tap water. Subsequently, they were washed several times with sterile water and dried quickly in a sampling bag containing silica gel. The samples were then stored at −80 °C until used.

### 4.2. DNA Extraction and High-Throughput Sequencing

Total DNA was extracted from frozen leaf samples by the modified cetyltrimethylammonium bromide (CTAB) method, and the quality of DNA was assessed by 1.5% agarose gel electrophoresis. The DNA was fragmented by mechanical interruption (ultrasound) and then purified, and end repaired. PolyA tails were added to the 3′ ends, and the fragments were ligated with sequencing adapters. The required fragment size was selected by agarose gel electrophoresis. PCR amplification was performed to form a sequencing library, and the qualified library was sequenced using the BGISEQ-500 platform with PE150 read lengths according to the manufacturer’s instructions. DNA extraction and sequencing were all performed by Guangzhou Bio&Data Biotechnologies Co., Ltd. (Guangzhou, China).

### 4.3. Chloroplast Genome Assembly and Annotation

At least 5 G of raw data were obtained for each species. After data filtering, adapter sequences and low-quality reads were removed to obtain high-quality clean data. First, NOVOPlasty software (k-mer = 39) was used for assembly and splicing, where the size of the insert was set to 250 bp [57]. Subsequently, the online program GeSeq was employed to annotate the chloroplast genome sequence, and Geneious v9.0.2 was used for visualizing the annotated sequence with manual corrections [58,59]. The annotated sequencing data for the four species were uploaded to the NCBI database with serial numbers ON707030, ON707031, ON707032, and ON707033, respectively. Finally, with the help of the online program Organellar Genome DRAW, the genome maps of the *Caladium* chloroplast genomes were constructed [60].

### 4.4. Comparative Analysis of Chloroplast Genomes

Relative synonymous codon usage (RSCU) analysis for every codon in each genome was conducted to determine codon bias [61]. The expansion and contraction of the IR borders of the four chloroplast genomes of *Caladium* were mapped with the aid of an IRscope [62]. Chloroplast genome similarity was assessed using BLAST Atlas on the GView server (http://server.gview.ca/, accessed on 26 October 2022) with 50 kbp connection windows with C. bicolor genome as a reference [63]. The Perl program provided by MIcroSAtellite Identification Tool (MISA) was used to analyze simple repeat sequence (SSR) sites. For mononucleotides, dinucleotides, trinucleotides, tetranucleotides, pentanucleotides, and hexanucleotides, the repetition thresholds were set to 10, 5, 4, 3, 3, and 3, respectively.

### 4.5. Phylogenetic Analysis

Published and fully annotated chloroplast genome-wide data of another eight species of Araceae were downloaded from the NCBI database, including *C. Bicolor* (NC_060474), *Syngonium angustatum* (MN046894), *Zomicarpella amazonica* (NC_051874), *Zamioculcas zamiifolia* (NC_048973), *Xanthosoma helleborifolium* (NC_051873), *Xanthosoma sagittifolium* (MW628970), *Pinellia pedatisecta* (NC_058756) and *Arisaema ringens* (NC_044118). A phylogenetic tree was constructed based on the complete chloroplast genome sequences of 12 species with maximum likelihood (ML) and Bayesian inference (BI) methods [64]. The ML tree was conducted using IQ-TREE v.2.1.4 [65]. The best-fitting nucleotide substitution model TVM+F+R3 was determined using the Bayesian Information Criterion (BIC) by ModelFinder in the IQ-TREE package, and 1000 bootstrap replicates [66]. The Bayesian inference was performed with MrBayes v.3.2.7, employing the GTR+G model of nucleotide substitution [67]. After the phylogenetic tree was exported, it was viewed using FigTree version 1.4.2.z.

## Figures and Tables

**Figure 1 genes-13-02180-f001:**
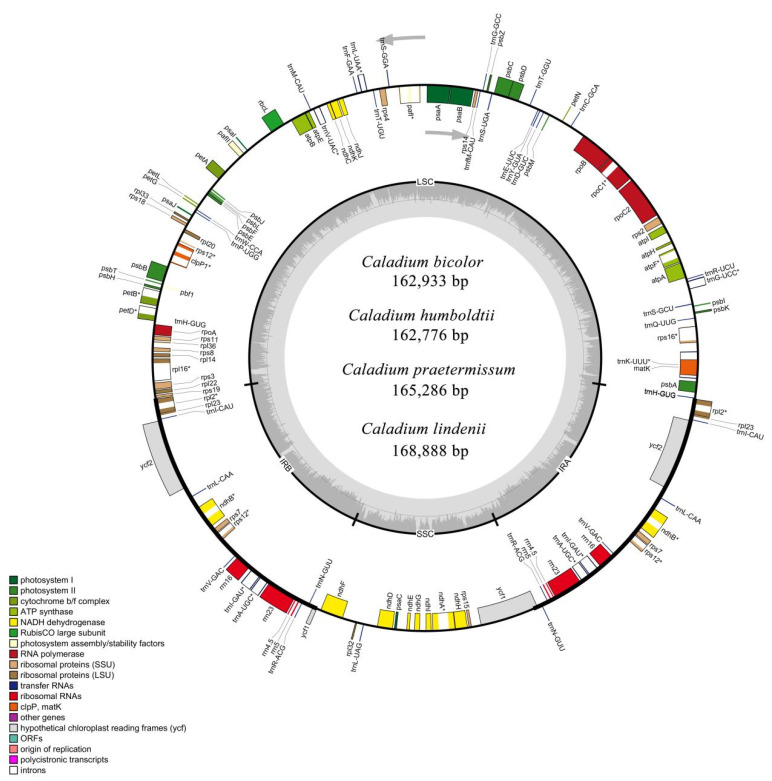
Circular gene map of the complete chloroplast genomes of four *Caladium* species.

**Figure 2 genes-13-02180-f002:**
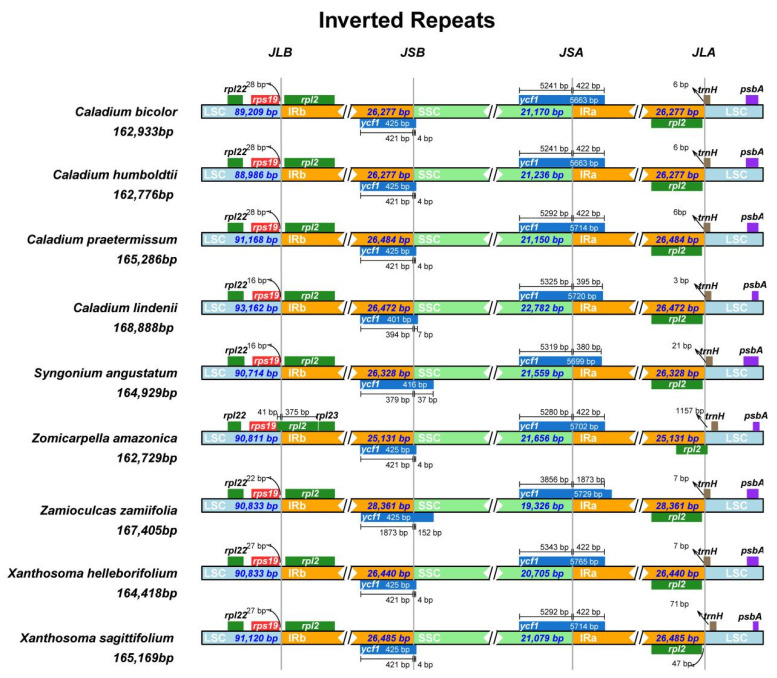
Comparison of the borders of the LSC, SSC, and IR regions of nine species of Araceae. The numbers above the gene features denote the distance between the gene borders, either the start or end of genes, and the junction sites.

**Figure 3 genes-13-02180-f003:**
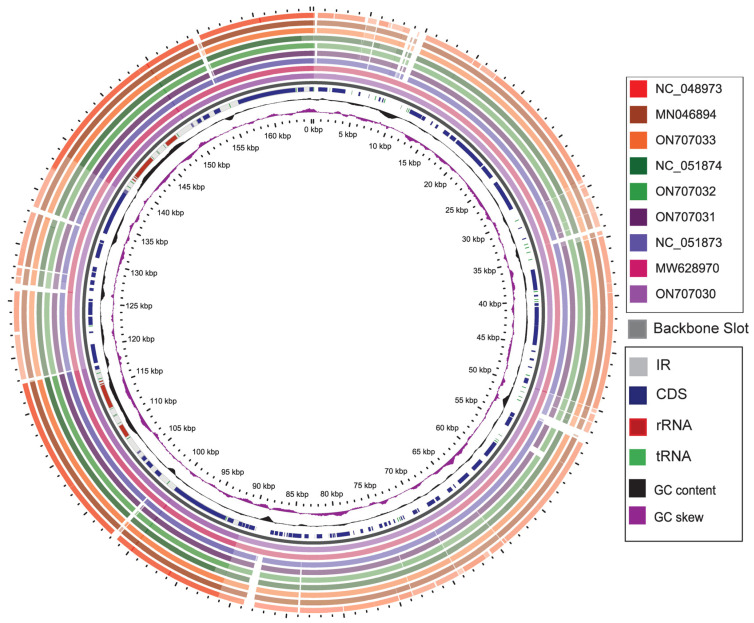
The graphical map shows nine circular plastome assemblies using *C. bicolor* plastome alignment as a reference. The innermost ring shows the genome size in kbp, followed by GC skew (purple) and GC content (black). From the inner to outer: *C. praetermissum* (ON707030), *X. sagittifolium* (MW628970), *X. helleborifolium* (NC_051873), *C. bicolor* (ON707031), *C. humboldtii* (ON707032), *Z. amazonica* (NC_051874), *C. lindenii* (ON707033), *S. angustatum* (MN046894), *Z. zamiifolia* (NC_048973). The similar and divergent locations are represented by continuous and interrupted track lines, respectively.

**Figure 4 genes-13-02180-f004:**
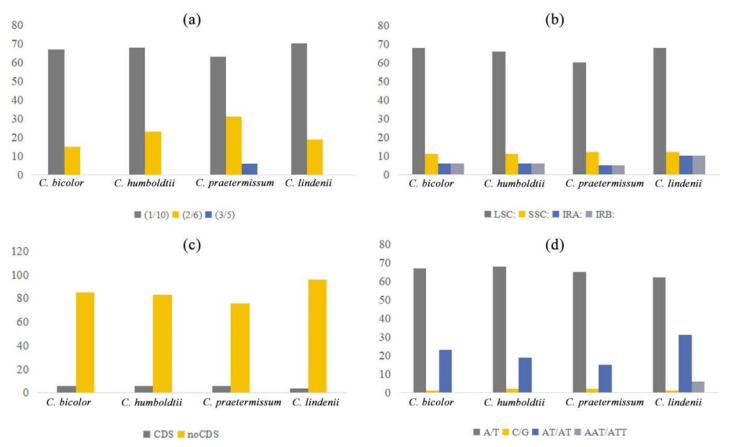
Comparison of repeat sequences in chloroplast genome distribution of four *Caladium* species. (**a**) The frequencies of different SSR repeat classes; (**b**) The SSR distribution in different chloroplast genome regions; (**c**) The SSR distribution in CDS and noCDS; (**d**) The SSR distributions of mono-, di-, and trinucleotide motifs.

**Figure 5 genes-13-02180-f005:**
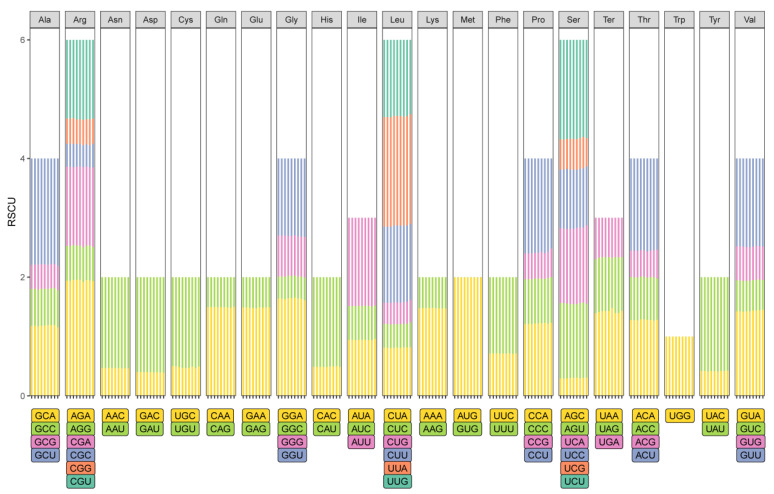
Relative synonymous codon usage (RSCU) analysis in chloroplast genes of *Caladium* species.

**Figure 6 genes-13-02180-f006:**
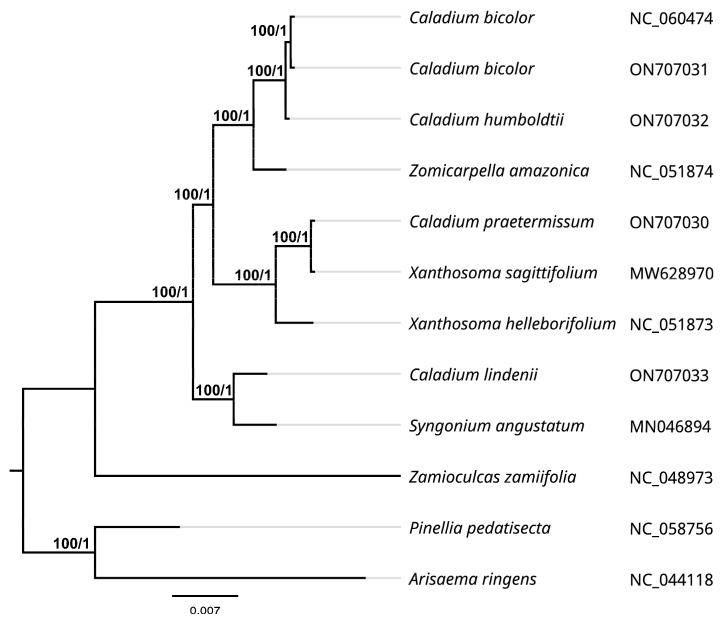
Maximum likelihood (ML) phylogenetic tree construction, including twelve species based on concatenated sequences from all chloroplast genomes. The obtained bootstrap values (BS) and Bayesian inference posterior probabilities (PP) are marked at the tree node (BS/PP).

**Figure 7 genes-13-02180-f007:**
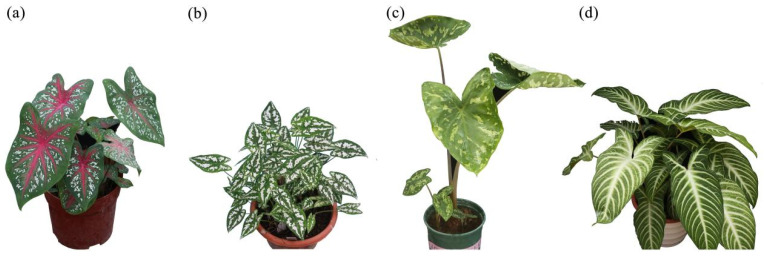
The four *Caladium* species used in the present study. (**a**) *C. bicolor*; (**b**) *C. humboldtii*; (**c**) *C. praetermissum*; (**d**) *C. lindenii*.

**Table 1 genes-13-02180-t001:** Comparison of the general information of nine species of Araceae.

Species Names	Size (bp)/GC Content (%)				Number of Gene			
	Genome	LSC	SSC	IR	PCGs	RNA/rRNA	RNA/tRNA	Total Genes
*C. bicolor*	162,933/35.87	89,209/34.17	21,170/29.07	26,277/41.5	86	37	8	131
*C. humboldtii*	162,776/35.91	88,986/34.26	21,236/28.99	26,277/41.5	86	37	8	131
*C. praetermissum*	165,286/35.64	91,168/33.76	21,150/29.27	26,484/41.42	86	37	8	131
*C. lindenii*	168,888/35.09	93,162/33.26	22,782/27.87	26,472/41.43	86	37	8	131
*Syngonium angustatum*	164,929/35.72	90,714/33.94	21,559/28.98	26,328/41.53	86	37	8	131
*Zomicarpella amazonica*	162,729/35.82	90,811/34.33	21,656/28.76	25,131/41.55	85	37	8	130
*Zamioculcas zamiifolia*	167,405/35.7	91,357/34.02	19,326/29.47	28,361/40.52	86	37	8	131
*Xanthosoma helleborifolium*	164,418/35.84	90,833/33.88	20,705/41.53	26,440/41.53	86	37	8	131
*Xanthosoma sagittifolium*	165,169/35.67	91,122/33.79	21,079/29.36	26,484/41.42	86	37	8	131

## Data Availability

The four-chloroplast genome sequence data generated in this study are available in GenBank of the National Center for Biotechnology Information (NCBI) under the access numbers: ON707030-ON707033.

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
