# Peer review of "Complete Chloroplast Genome Sequences of Four Species in the Caladium Genus: Comparative and Phylogenetic Analyses"

_genes, 2022, doi:10.3390/genes13122180_

Round 1

Reviewer 1 Report

This manuscript describes the chloroplast genome sequences of four different Caladium species that are well-known ornamental plants. The workflow is accurate and the results are well-described. Eventually, there is a need to polish the grammar. Additionally, there are few (not confined to) comments I need to raise up to improve the write-up of this work.

Introduction

L60 full name of AFLP with abbreviation in parenthesis.

L62 full name of SSR with abbreviation in parenthesis

L184-187 Please describe how different and how conserve they are.

L196 What are GC1-3? I cannot find any information on that in the text. All results should be clearly described even when the readers did not refer to figures or tables.

Results

L94-87 I do not see any point to use abbreviations for the four species used in this study. I highly suggest the authors to just list them out accordingly.

L134-136 this is a discussion material.

L149 what type of inconsistencies?

L262 There is no discussion on the phylogenetic analysis. I am aware that there is limited finding for phylogenetics using gene markers; however, you can discuss on the molecular placements of Caladium in Araceae, or relate them with SSR or AFLP.

Materials and methods

L276 full name of CTAB

L295 please put the accession numbers in ascending orders.

L294 genome maps

L300 globally? I don't understand

L300 Shuffle-LAGAN. Also, I did not see any results of mVISTA in this work. and where is GView methodology?

L312 In the Results, the authors mentioned they have performed both ML and BI analysis; BI was not mentioned in Materials and methods. Then, information in Figure 6 is severely lacking. The substitution model was not mentioned for ML. No citation was provided for the programs used.

Figure 1 Spelling mistake in the caption for "chloroplast"

Figure 3 please list out the species names in the legend and caption. The figure should be self-explanatory.

Figure 6 I am quite confused with the ML tree constructed. It looks like it is a consensus tree showing the topology, but there is a scale bar at the bottom. Please clarify or fix it. Also, the bootstrap support is incorrect. Plus, the authors mentioned they perform BI analysis too' where is the result? Then, the tree is constructed using concatenated sequence from all chloroplast genome? What sequence? You mean CDS?

References

There are several references that are wrongly written in this section, for example 4. The names of the authors are incorrectly assigned. I would refer this as a disrespect to the authors too. Please check throughout and fix them accordingly.

For reference 64, the authors mentioned RAML (I assume it should be RAxML), but the reference is on IQ-TREE. Please check everything and fix them accordingly.

Reviewer 2 Report

Ye et al. sequenced, assembled, and annotated the complete chloroplast genomes of four species in the Caladium genus (Caladium bicolor, Caladium humboldtii, caladium praetermissum, and Caladium lindenii) and compared them with other chloroplast genomes. The study is fine, and the results might be helpful for evolutionary studies. However, I would suggest some necessary revisions as follows:

1- I would suggest changing the manuscript title. See below one suggested title:

"Complete chloroplast genome sequences of four species in the Caladium genus: comparative and phylogenetic analyses".

2- The complete chloroplast genome of Caladium bicolor was previously submitted with accession number "NC_060474". It is already there! I am wondering why the authors did not use this genome. Can you explain this? Also, the authors should add a section to conduct a comprehensive comparison between NC_060474 and ON707031.

3- The four chloroplast genomes of Caladium species with accession numbers ON707031, ON707032, ON707030, and ON70703, are not publically available. I could not review and replicate the phylogenetic tree and other results generated in this study without having the chloroplast genomes.  

Reviewer 3 Report

1- Add some information about high-throughput sequencing and genome sequencing methods in the Introduction.
2- Arrange references based on the journal format.
3-English language needs revision.

Round 2

Reviewer 2 Report

Thank you for addressing my comments. The current manuscript has been much improved.

Some minor edits as follows are required:

1- In lines 10-11, I suggest mentioning the full names of four Caladium species in the abstract.  

2- In line 11, I suggest adding an abbreviation for the chloroplast (cp).

3- In Table 1: Please replace word accessions with "species names". In the last column "Total", the authors should clarify the total of what?

4- Can you double-check the tree in figure 6 and increase the quality?

Author Response

Point 1: In lines 10-11, I suggest mentioning the full names of four Caladium species in the abstract.  

Response 1: We appreciate your suggestion and have added the full names of four Caladium species in the revised manuscript.

Point 2: In line 11, I suggest adding an abbreviation for the chloroplast (cp).

Response 2: We appreciate your suggestion and have added an abbreviation for the chloroplast (cp) in the revised manuscript.

Point 3: In Table 1: Please replace word accessions with "species names". In the last column "Total", the authors should clarify the total of what?

Response 3: We appreciate your suggestion and have revised the relevant information in Table 1.

Point 4: Can you double-check the tree in figure 6 and increase the quality?

Response 4: We appreciate your suggestion. We have checked the tree in Figure 6 and added relevant description.